# Investigation into PVDF-HFP and PVP Polymer Blend Electrolytes with Lithium Ions for Energy Storage Application

**DOI:** 10.3390/polym17131758

**Published:** 2025-06-25

**Authors:** Bilash Jyoti Gogoi, M. Murugesan, N. Nallamuthu, P. Devendran, Arumugam Murugan, Radak Blange, Muthaiah Shellaiah

**Affiliations:** 1Department of Chemistry, North Eastern Regional Institute of Science & Technology (NERIST), Itanagar 791109, Arunachal Pradesh, India; 518014@nerist.ac.in; 2Department of Physics, School of Advanced Sciences, Kalasalingam Academy of Research and Education, Krishnankoil 626126, India; m.murugesan@klu.ac.in; 3Department of Physics, Dayananda Sagar of Academy of Technology and Management, Bangalore 560082, India; nallamuthu-phy@dsatm.edu.in (N.N.); devendran-phy@dsatm.edu.in (P.D.); 4Department of Electrical Engineering, North Eastern Regional Institute of Science & Technology (NERIST), Itanagar 791109, Arunachal Pradesh, India; rb@nerist.ac.in; 5National Research Institute of Chinese Medicine, Ministry of Health and Welfare, Taipei 11221, Taiwan

**Keywords:** ionic conductivity, PVDF-HFP/PVP, polymer blend electrolyte, XRD, FTIR, lithium iodide

## Abstract

To improve solid-state lithium batteries, solution casting has been employed to create lithium ion-conducting copolymer electrolytes involving poly(vinylidene fluoride-co-hexafluoropropylene) (PVDF-HFP)/polyvinylpyrrolidone (PVP) blend polymers with various compositions. Following X-ray diffraction and Fourier transformation (FTIR), the structural characterisation and identification of molecular bonding in polymer electrolytes were confirmed. Through AC impedance analysis, the electrical characteristics of the solid-state polymer films were investigated. The dielectric conductivity of the sample was found to obey the modified Arrhenius relationship, while in the case of a sample with higher conductivity, it followed Arrhenius behaviour. The relaxation parameters and dielectric behaviour of the samples are demonstrated and discussed.

## 1. Introduction

The cost of electrical energy devices has increased worldwide even though there are plenty of energy resource devices on the market. The requirement for energy storage is necessary to overcome the shortages of conventional energy sources. This research is focused on the development of solid-state batteries and supercapacitors to store energy. Recently, Li-ion batteries have performed and sold better than other batteries like Ni-Cd, Pb-acid, Ni-MH, etc. The solid-state lithium-ion battery provides better enhancement of electrochemical performance. Lithium, the lightest of all metals, is the main reason for using an anode, and lithium salt-based polymer electrolytes exhibit a wide electropositive potential window. The main drawbacks associated with batteries based on liquid/aqueous electrolytes are limited operational temperature, electrode corrosion, difficulties in hermetic sealing, and the enrichment of metal dendrites through the electrolyte medium during cycling that may be encountered. These leading drawbacks in batteries can be rectified through the use of solid polymer electrolytes. Owing to its stable dimensions, safety, flexibility, processability, electrochemical stability, and longer life, a solid-state polymer electrolyte should be used in energy storage devices [1]. With 20 wt.% AgNO_3_ doping, polymer electrolyte films made using polyvinyl alcohol have demonstrated maximal conductivity [2]. The higher level of amorphicity may be the cause of the increased conductivity seen in electrolytes based on PVDF-HFP/PEMA [3]. Polymer blend electrolytes with KIO_3_ offer fascinating alternatives to diverse solid-state batteries [4]. Ionic conductivity increased significantly in thin films made from a PVC-PMMA mixture containing lithium triflate salt (LiCF_3_SO_3_) with a dibutyl phthalate plasticizer [5]. With increasing LiBF_4_ concentration in DMF solution, the FTIR spectra of PVDF with a dimethyl formamide (DMF) plasticiser mixed with LiBF_4_ underwent a shift in IR for the O=C-N band stretches. These O=C-N bands appearing in the DMF spectrum and in the Li^+^ cations interact strongly, as indicated by the shift in the peak shape, bandwidth, and relative strength and the emergence of an additional band at this wavenumber [6]. Several redox agents, such as KI, LiI, and NaI, have been used to begin making PVDF-HEP/PMMA nanocomposite polymer blend membranes (TBAI) [7]. For membranes formed from gel polymer blends based on PVDF-HFP:PMMA-(PC:DEC) LiClO_4_, conductivity improved when LiClO_4_ salt was added. This increase in the amorphicity of the polymer gel blend leads to an increase in the ionic conductivity until the salt content reaches 7.5 wt.% LiClO_4_. These electrolytes appear to be ideal for electrochemical process-based devices, such as lithium-ion batteries, due to their high ionic conductivity or transference number (around unity) [8]. When the ratio of PVP to PVDF-HFP to the liquid electrolyte was chosen with a weight ratio of 1:10:18, based on the mixture of the two polymers, the polymer electrolyte demonstrated a conductivity of 0.4 Scm^−1^ at ambient temperature [9]. Through PVDF’s reduced degree of crystallinity and the addition of 7.5 weight percent of PMMA, the PVDF-LiClO_4_-DMP matrix’s conductivity can be improved [10]. The PVDF-HFP membrane’s thermal stability was significantly improved through doping with Octavinyl-Polyhedral Oligomeric Silsesquioxane (OVAPOSS), avoiding shrinking at a temperature of around 160 °C and retaining a specific amount of strength with the increasing thermal stability of the battery [11]. At 363 K, it was discovered that the combination of 40 wt.% LiBF_4_ salt with a PVDF-HFP polymer exhibits the best ionic conduction [12]. Ionic conduction through the PVDF-HFP/HDPE membrane has displayed excellent electrochemical characteristics for lithium-ion batteries [13]. Al_2_O_3_ cross-linked with a PVDF-HFP porous polymer membrane demonstrated how cross-linked networks can reinforce mechanical performance, while the application of pore-forming chemicals significantly boosted flexibility and electrolyte uptake [14]. Suggestions for the formulation of composite electrolytes for complete solid-state batteries with lithium ions will provide fundamental insights into interstitial Li^+^ ion conduction and chemical bonding across polymer/ceramic interfaces [15]. Even though PVdF-based solid electrolytes have good electrochemical properties, they are unstable with lithium and lithium salts because of the low interfacial properties between fluorine and lithium. PVdF-HFP and its copolymer exhibit excellent solubility. Here, we aim to develop lithium-based PVDF-HFP–PVP solid polymer electrolytes in which lithium iodide exists in salts with different compositions. Lithium iodide has a large ionic radius and produces a large lattice gap to enhance ionic migration between polymer chains. Hence, the mixture of the two polymers PVP and PVDF-HFP, with various ratios of lithium iodide salts, was chosen for the current study due to their optimal electrical properties. This polymer blend electrolyte was synthesised through the solution casting method, and its viability was examined using several characterisation methods and conductivity measurements.

## 2. Materials and Methods

### 2.1. Materials

Precursors, such as polymers of PVDF-HFP (molecular weight = 300,000 g/mol), PVP (molecular weight = 90,000 g/mol), and LiI, were procured from Sigma Aldrich, Bangalore, India. A DMF plasticiser was obtained from Merck, Maharashtra, India.

### 2.2. Preparation of PVDF-HFP/PVP/LiI-Based Solid Polymer Electrolyte

Solid polymer electrolytes based on PVDF-HFP/PVP/LiI were synthesised using specific concentrations of PVDF-HFP (60 wt.%), PVP (40 wt.%), and lithium iodide (5, 10, 15, and 20 wt.%), denoted as PPHLI5 wt.%, PPHLI10 wt.%, PPHLI15 wt.%, and PPHLI20 wt.%, respectively. To achieve a homogeneous and transparent solution, the required amounts of PVP, PVDF-HFP, and lithium iodide were individually dissolved in DMF and agitated for six hours. The solutions were then combined and continuously stirred for another six hours to induce gelation, followed by casting into Petri dishes. The resulting polymer films exhibited highly uniform surfaces, allowing for further characterisation and conductivity analysis.

### 2.3. Characterization of Solid Polymer Electrolyte Films

A synthesised solid polymer electrolyte film of PPHLI5 wt.%, PPHLI10 wt.%, PPHLI15 wt.%, and PPHLI20 wt.% was investigated using different characterisation methods. The X-ray diffraction pattern of the prepared solid polymer film was captured through a Bruker X-ray diffractometer at an X-ray wavelength of 1.540 Å. The solid polymer film’s scanning range was set from 10° to 80°. The Fourier-transform infrared (FTIR) transmittance spectra of the prepared films were captured via the SHIMADZU IR Tracer-100 spectrometer (Shimadzu Corporation, Kyoto,Japan) for wavenumbers ranging from 4000 cm^−1^ to 400 cm^−1^. The impedance measurement of the polymer electrolyte was measured through the computer-controllable HIOKI 3532-50 LCR Hi-Tester (Hioki, Ueda, Japan) for frequencies varying from 42 Hz to 1 MHz for a temperature span of 303–373 K.

## 3. Result and Discussion

### 3.1. XRD Analysis

Figure 1 displays the XRD patterns of PPHLI5 wt.%, PPHLI10 wt.%, PPHLI15 wt.%, and PPHLI20 wt.% polymers. The obtained XRD patterns reveal pure PVDF-HFP, PVP, and lithium iodide contents. From Figure 1, we can see that the typical peaks for PVDF-HFP show up at 20°, indicating that the PVDF polymer is present in a solid crystal form [16]. Further, characteristic peaks at 30° and 40° are detected. The peak at 30° shows the pure amorphous nature exhibited by the PVDF-HFP polymer. This indicates the reduction in the crystallinity of the solid polymer electrolytes with the inclusion of plasticisers and lithium iodide. Moreover, the XRD pattern does not reveal any characteristic peaks for lithium iodide. This signifies the well-complexed lithium salts in the polymer matrix, which validates the complexation of the film. Such a highly amorphous nature leads to the improved ionic conductivity of the polymer electrolytes. From the obtained data, fixed complexation between the PVDF-HFP, the plasticiser, the PVP polymers, and the lithium iodide [17] in the resultant polymer film is confirmed.

### 3.2. FTIR Studies

The Fourier-transform infrared spectrum (FTIR) is an essential record that provides detailed information throughout the polymer structure. The FTIR spectrum is used to explain the polymerisation mechanism of PVDF-HFP, PVP, and lithium iodide. The FTIR spectra obtained for the synthesised polymer electrolyte films of PVDF-HFP, PVP, and lithium iodide salts and their compounds are presented in Figure 2 and Figure 3. The FTIR transmittance spectra for the PPHLI5 wt.%, PPHLI10 wt.%, PPHLI15 wt.%, and PPHLI20 wt.% polymers are presented. The peaks obtained at 1047 cm^−1^, 1188 cm^−1^, and 1656 cm^−1^ were attributed to the CF_3_ out-of-plane deformation of PVDF-HFP, CF_2_ antisymmetric stretching in PVDF-HFP, and a vibration band that corresponds to C=O stretching, respectively [18]. The peaks at 1281 cm^−1^, 875 cm^−1^, and 813 cm^−1^ were attributed to α-phase formation at the PVDF-HFP crystalline peak [19]. The peak at 1422 cm^−1^ corresponds to the CH_2_ wagging vibrations of the bending polymer (PVDF-HFP/PVP polymer). The peak at 3469 cm^−1^ represents OH stretching in PVP, while blending PVDF-HFP/PVP with various concentrations of lithium iodide results in a fluctuating peak location, shape, and intensity. The –CH_3_ asymmetric stretching in PVDF is assigned to the 2938 cm^−1^ peak [20]. When the lithium iodide salts accumulated at the synthesised polymer, the vibrational state band in the region of 760 cm^−1^ resembled the crystallinity of PVDF-HFP, which shifted to a lower wavenumber by 741 cm^−1^, with a decrease in intensity due to the increased concentration of lithium iodide. Thus, the obtained data reveals the characteristics of the amorphous polymers, which are primarily due to salt dispersion throughout the polymer matrix. The vibrational state band that appears at 879 cm^−1^ shifts to 877 cm^−1^ due to the lithium iodide salt dispersion, which reveals the complexation and presence of the salt throughout the polymeric host matrix [5,21].

### 3.3. AC Impedance Studies

The Nyquist plots obtained for the PPHLI5 wt.%, PPHLI10 wt.%, PPHLI15 wt.%, and PPHLI20 wt.% polymers at room temperature are shown in Figure 4. For all the solid polymer electrolytes, there exists a depressed semicircular portion. When increasing the salt content, a spike appears in the complex impedance representation. In comparison to a semicircle with a spike that corresponds to the resistance in a parallel connection and the capacitance in a series connection, the semicircles are thought to represent equivalent electrical circuits with parallel connections for both resistance and capacitance. This semicircle is observed to be large. The linear area, which, in the lower frequency range, characterises the impact of the stopping electrodes and the frequency region, describes the bulk effects in the electrolytes. The real axis (Z’) intercept obtained for the semicircle provides the electrolytic resistance (*R_b_*). Bulk resistance was measured using Z-View software 3.5i version and an equivalent circuit fit. The ionic conductivity can be determined using the following relationship:(1)$$\sigma =\frac{l}{R_{b}A}$$
where ‘l’ is the thickness of the polymer electrolyte film, ‘*R_b_*’ is the bulk resistance found from the *x*-axis point in the complex impedance plot of the AC circuit, and ‘*A*’ is the surface area of the polymer film. As the lithium iodide concentration increases, the ionic conductivity through the solid polymer electrolyte increases.

Figure 4 illustrates the better conductivity of PPHLI15 wt.% lithium iodide at room temperature when compared to other concentrations. The resistance of the electrolyte is found to decrease with an increase in temperature. This decrease in resistance appears due to an increase in ionic mobility and the increase in the number of ionic carriers with the temperature of the polymer electrolyte [22,23,24]. Figure 5 displays the PPHLI15 wt.% impedance spectra for different temperatures of lithium iodide salt. The semicircle diameter obtained is found to be reduced with an increase in thermal energy, which represents the improved bulk conductivity with rising temperature [17].

### 3.4. Measurements of Conductance Spectra

Figure 6 displays the conductance spectra for the PVDF-HFP/PVP electrolyte with different concentrations of lithium iodide salt. The spectra support the presence of two separate zones, a dispersion region in a higher frequency range and a plateau in a lower frequency range, for the samples prepared with different compositions. The higher frequency dispersion zone accounts for the phenomenon of bulk relaxation, while the middle region is comparable to the frequency-independent region (the plateau portion). The DC conductivity is determined via extending the curve along the *y*-axis direction in the low-frequency range. From Figure 6, it can be distinguished that σ_dc_ increases with increasing lithium iodide concentration [25]. The highest conductivity at room temperature is obtained for the PPHLI15 wt.% polymer electrolyte compared to the other prepared electrolytes. The polarisation due to the effect of charge transfer, which manifests at the electrode–electrode interface along the long range of hopping ions, causes the conductivity to be extremely low in the lower frequency area for all samples [26]. Due to the ions’ swinging, the conductivity increases linearly when frequency is increased in an area of higher frequency dispersion. Figure 7 shows the conductivity spectra of the sample with the highest conductivity, namely PPHLI15 wt%, at several temperatures. The plasticizing effect of PVP led to a significant increase in ionic conductivity, attributed to improved ionic mobility and an increased concentration of charge carriers [27,28,29].

### 3.5. Temperature-Dependent Conductivity Studies

The activation energy of the sample can be determined using the following relation:σ = (σ_0_) exp (−E_a_/kT)(2)

As the temperature rises, the structural relaxation in the polymer and the ionic behaviour at the coordination sites result in increased electrical conductivity [30,31,32,33,34]. The Arrhenius relationship holds for highly conducting polymer electrolytes. The conductivity, which is independent of temperature as well as the dielectric constant, is verified from the modified Arrhenius correlation (Figure 8).

### 3.6. Dielectric Constant Studies

Dielectric loss can be defined as the measurement of the energy required to transfer ions with an abrupt reversal in the polarity of an applied electric field, whereas the dielectric constant represents the quantity of charge stored in a substance [35]. The system’s complex permittivity (ε*), often known as its dielectric constant, can be expressed asε* = ε′(ω) − i ε″(ω)(3)
where the real part ε′ and the imaginary part ε″ indicate the energy storage and loss observed for each cycle of the electric field applied, respectively [36,37].

Figure 9a,b display how the dielectric constant (ε′) and the dielectric loss (ε″) vary over time, respectively. The dielectric constant (ε′) value is seen in Figure 9a to be quite high at low frequencies while decreasing with increasing frequency and remaining roughly constant over higher frequency values. The reason for this higher ε′ value may be due to space charge effects, which result in the buildup of charge carriers near the electrodes [38]. The fast periodic reversal of the electric field may be the cause of the drop in ε′ at high frequencies. As a result, the electrode–electrolyte interface’s polarisation caused by charge accumulation decreases [39]. In addition, the polymer chain becomes more flexible at higher temperatures, which further lessens PVDF’s tendency to crystalline. Beyond PPHLI15 wt.%, the systems exhibit a reduced dielectric constant at lower frequencies due to the formation of extensive ion clusters, which restrict ion mobility and diminish effective conduction pathways. This aggregation effect hinders charge transport efficiency, impacting overall electrolyte performance.

### 3.7. Real and Imaginary Modulus Studies

Applying the electric modulus spectra can shed light on how ions move across ionic networks. Figure 10 displays the modulus spectra of the greater conductivity in the electrolytic polymer. It also shows the broad relaxation over a range of frequencies when the expected M’ value is increased. The modulus values are 0 at low frequencies, which is intended to account for the small effect of electrode polarisation [38,39]. There is a peak that appears in the imaginary region of the modulus spectrum, suggesting that segmental movement and ionic conductivity are connected and cause this peak [40,41]. Long-range ionic mobility in the polymer is what gives rise to the peak that is observed in the M” curve. With increasing salt concentration, this peak shifts to a higher frequency region, as shown in the figure, demonstrating the thermal relaxation process.

### 3.8. Tangent Studies

A Tan δ curve recorded versus frequency was used to determine the dielectric relaxation parameter in the copolymer complexes PVDF-HFP, PVP, and LiI. The dielectric loss factor (Tan δ) is defined by(4)$$\delta =\frac{\varepsilon^{II}}{\varepsilon^{I}}\;\;$$

Figure 11 shows the fluctuation in Tan δ as a function of frequency for all copolymer complexes of PVDF-HFP, PVP, and LiI at 303 K. Tan δ was found to increase with increasing frequency for all the copolymer complexes, reaching a maximum and then decreasing with additional increases in frequency, as shown in Figure 11. The peak of the loss tangent is defined by the following relation for the highest dielectric loss:ωτ = 1(5)
where the applied electric field’s angular frequency and relaxation duration, respectively, are denoted by τ and ω. This reduction in relaxation time increased the ionic conductivity; it should be highlighted that PPHLI15 wt.% had the lowest relaxation time and therefore the maximum ionic conductivity [27]. Space polarisation was indicated by the dispersion that took place in the lower frequency range. The large likelihood of ionic jumps per unit time is what causes the high-intensity peak [26].

### 3.9. Argand Plot Studies

By analysing the Argand plot at various temperatures, the relaxation processes of the electrolytic polymers can be demonstrated. Figure 12 shows the Argand plot as a function of varying temperature, which describes the higher conductivity of the electrolytic polymer. This clearly demonstrates that the Argand plot curves at various temperatures form incomplete semicircles, indicating non-Debye behaviour. Multiple polarisation and relaxation mechanisms, as well as several interactions occurring between the dipoles and ions, all contribute to non-Debye relaxations [42,43].

In the PVDF-HFP/PVP system, dielectric behaviour is influenced by the intrinsic polarity of the polymers. PVDF-HFP possesses a high dielectric constant due to the presence of highly polarised CF_2_ dipoles, which enhance salt dissociation and increase the number of free charge carriers. Meanwhile, PVP, with its carbonyl (C=O) groups, contributes to dipolar orientation and Li^+^ ion coordination, aiding in polarisation and local field formation. This dual contribution leads to a substantial increase in dielectric permittivity at low frequencies, which is typically associated with electrode polarisation and space charge accumulation, key factors in boosting ionic conductivity. Furthermore, the observed non-Debye relaxation behaviour indicated by a deviation from ideal single-time-constant dielectric relaxation suggests a broad distribution of relaxation times. This pattern is characteristic of disordered or amorphous systems, like our blended polymer matrix, and indicates the coexistence of multiple ionic environments and dynamic coordination states. Such behaviour supports segmental motion and hop-based ion transport mechanisms, which are inherently non-uniform and thermally activated [39,40]. The polymer electrolyte’s conductivity and arc radius are closely related [44,45].

Figure 12 clearly shows the temperature data. As the temperature rises, the arc’s length decreases, confirming that the conductivity has increased.

A comprehensive evaluation of key parameters such as the conductivity, activation energy, dielectric constant, and relaxation time across various PVDF-HFP/PVP-LiI polymer blend formulations is systematically presented in Table 1. It can be observed that the ionic conductivity increases with increasing lithium iodide content up to an optimal concentration (PPHLI15 wt.%), which exhibits the highest conductivity of 7.504 × 10^−5^ S·cm^−1^. Correspondingly, the activation energy decreases, indicating enhanced ion transport and a reduced energy barrier for ionic conduction. The dielectric constant also shows a significant increase for this composition, which suggests improved charge carrier density and enhanced dielectric polarisation. Furthermore, the relaxation time decreases, implying faster ion mobility and improved charge relaxation dynamics. These combined factors confirm that the sample with 15 wt.% lithium iodide demonstrates superior electrochemical performance, making it the most suitable composition for potential applications in solid polymer electrolytes.

## 4. Conclusions

For the further development of lithium ion-based batteries, a solution casting approach has been used to synthesise a composite electrolytic polymer blend with lithium iodide and a PVDF-HFP/PVP polymer. Different characterisation techniques have been used to analyse structural and electrical behaviour. XRD analysis verified the salt’s and the polymer’s complex structural makeup. The amorphous characteristic of the prepared solid polymeric electrolytes was improved via the addition of lithium iodide up to 15 wt.%. The polymer matrix’s functional group was confirmed through FTIR measurement, and the modest variations in peak location support the interactions between the PVDF-HFP/PVP polymer and lithium iodide. The prepared lithium iodide-doped PVDF-HFP/PVP polymeric electrolyte blend with PPHLI15 wt.% had a better maximum conductivity of 7.504 × 10^−5^ S·cm^−1^. The Arrhenius relationship holds for more highly conducting polymer electrolytes. The conductivity, which is independent of temperature as well as the dielectric constant, was verified through the modified Arrhenius correlation. Ionic conductivity is higher in polymer matrices of solids with a greater dielectric constant, according to the dielectric analysis. This polymer electrolyte can be employed in battery applications as an electrolyte due to the lithium iodide mixed-blend polymer’s intermediate energy band gap and strong ionic conduction.

## Figures and Tables

**Figure 1 polymers-17-01758-f001:**
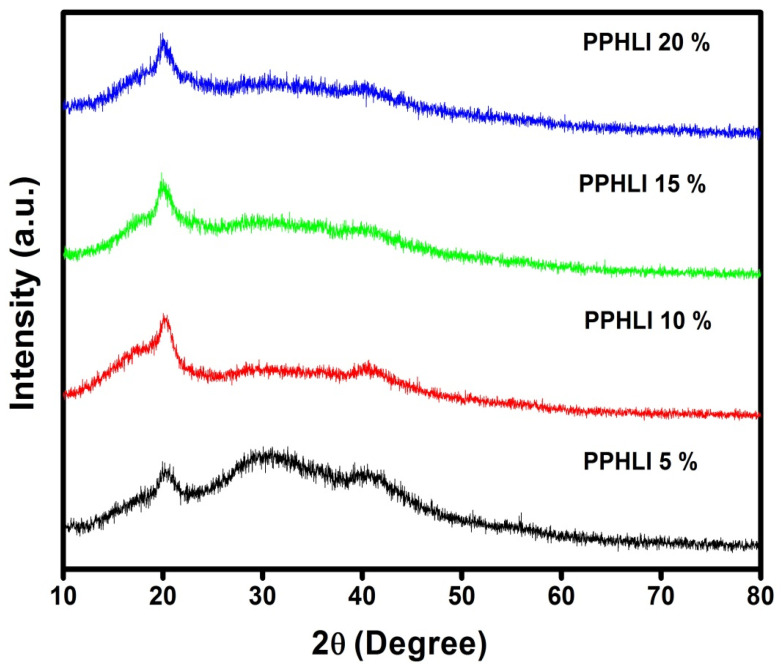
XRD patterns of PPHLI5 wt.%, PPHLI10 wt.%, PPHLI15 wt.%, and PPHLI20 wt.% polymers.

**Figure 2 polymers-17-01758-f002:**
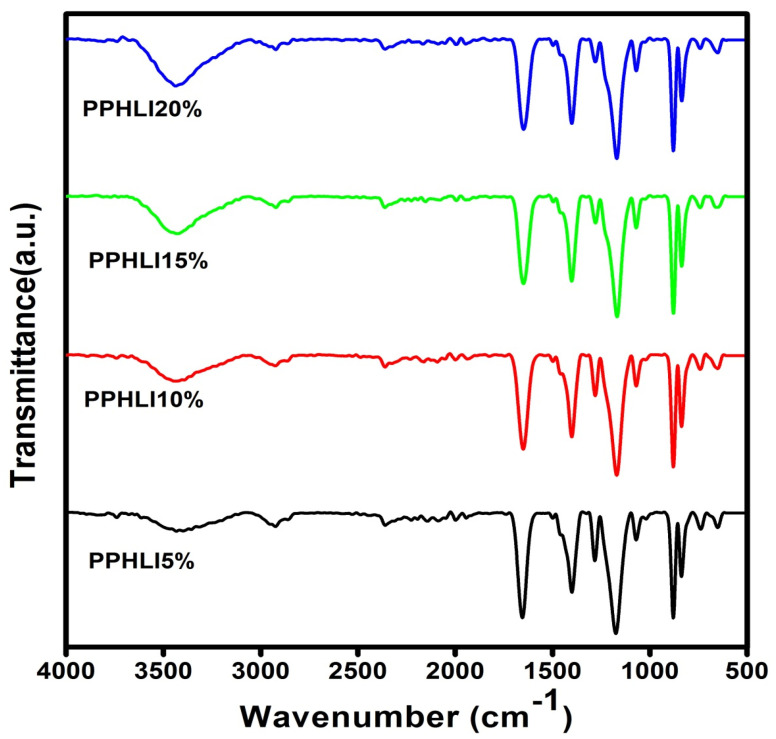
FTIR transmittance spectra for PPHLI5 wt.%, PPHLI10 wt.%, PPHLI15 wt.%, and PPHLI20 wt.% polymers.

**Figure 3 polymers-17-01758-f003:**
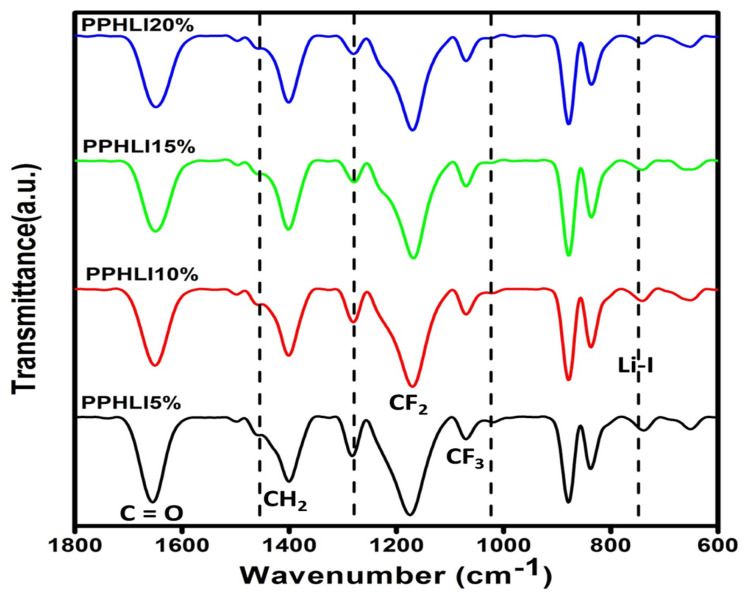
FTIR transmittance spectra (1800 cm^−1^–600 cm^−1^) for PPHLI5 wt.%, PPHLI5 wt.%, PPHLI10 wt.%, PPHLI15 wt.%, and PPHLI20 wt.% polymers.

**Figure 4 polymers-17-01758-f004:**
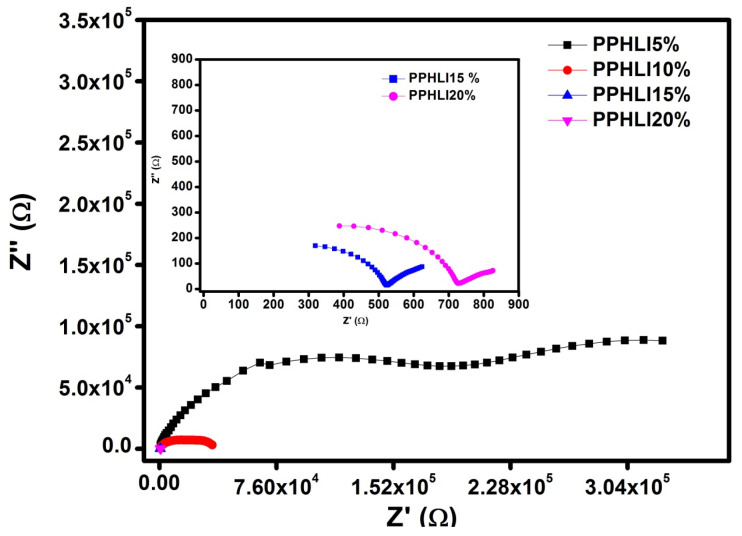
Cole–Cole plots of PPHLI5 wt.%, PPHLI10 wt.%, PPHLI15 wt.% and PPHLI20 wt.% polymers.

**Figure 5 polymers-17-01758-f005:**
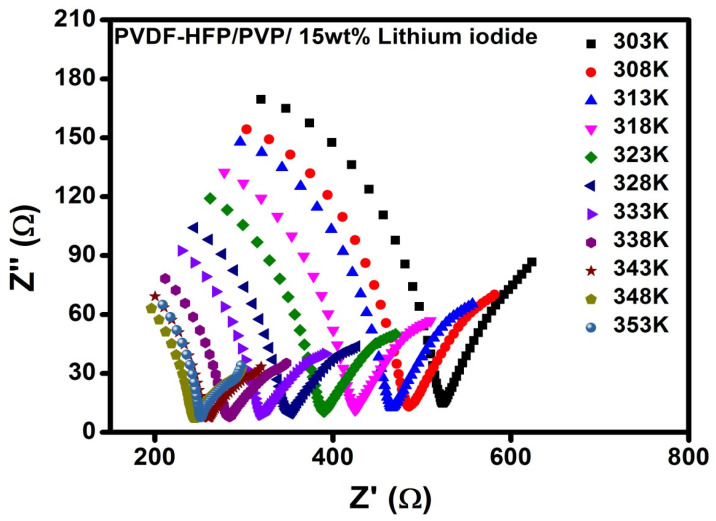
Impedance spectra for PPHLI15 wt.%, heat-treated at different temperatures.

**Figure 6 polymers-17-01758-f006:**
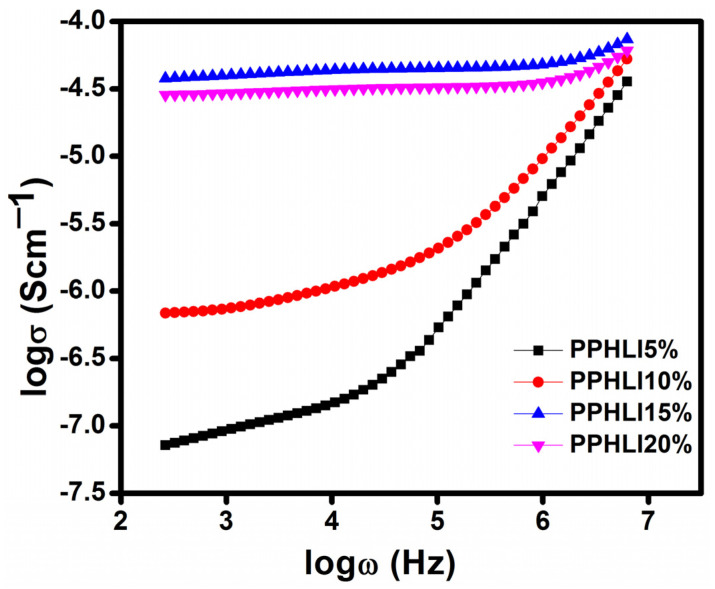
Conductance spectra for PPHLI5 wt.%, PPHLI10 wt.%, PPHLI15 wt.%, and PPHLI20 wt.% polymers.

**Figure 7 polymers-17-01758-f007:**
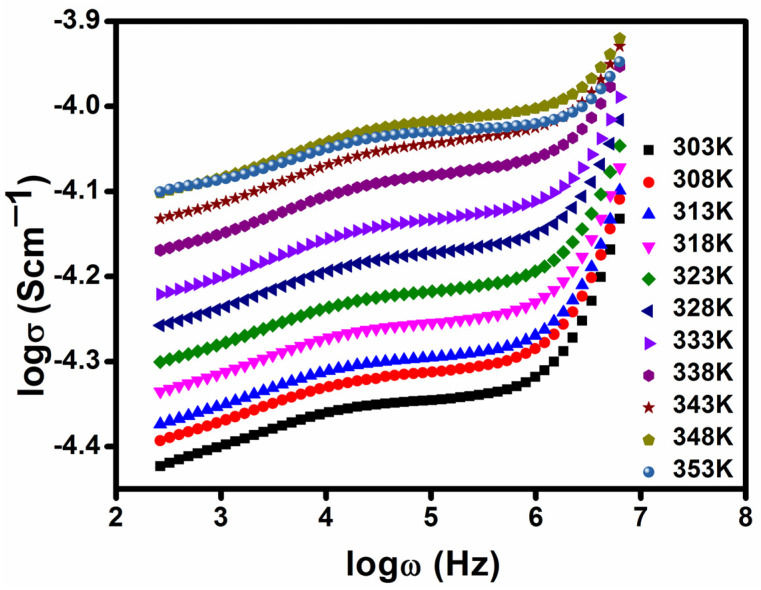
Conductance spectra of PPHLI15 wt.%, heat-treated at different temperatures.

**Figure 8 polymers-17-01758-f008:**
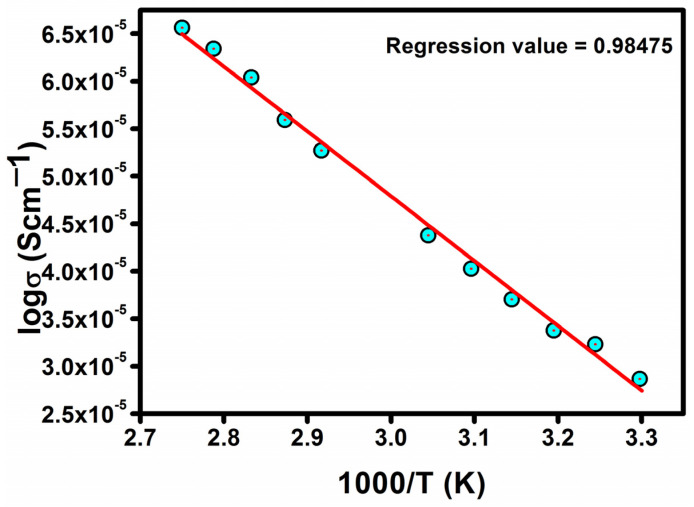
Arrhenius relationship for the PPHLI15 wt.% blend polymer electrolyte.

**Figure 9 polymers-17-01758-f009:**
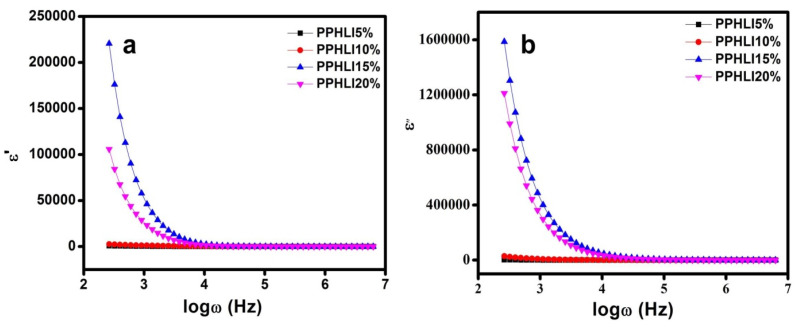
(**a**,**b**). Dielectric constant and dielectric loss of PPHLI5 wt.%, PPHLI10 wt.%, PPHLI15 wt.%, and PPHLI20 wt.% polymers.

**Figure 10 polymers-17-01758-f010:**
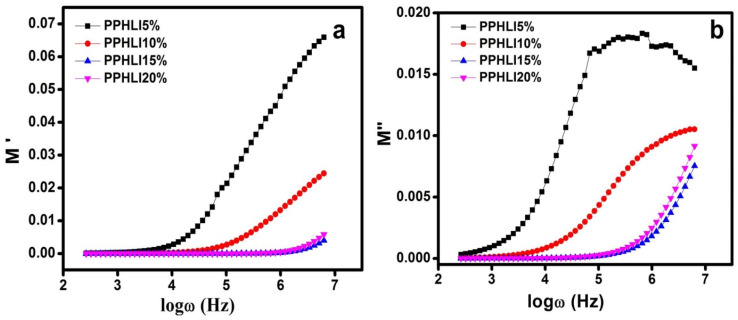
(**a**,**b**). Real and imaginary moduli of PPHLI5 wt.%, PPHLI10 wt.%, PPHLI15 wt.%, and PPHLI20 wt.% polymers.

**Figure 11 polymers-17-01758-f011:**
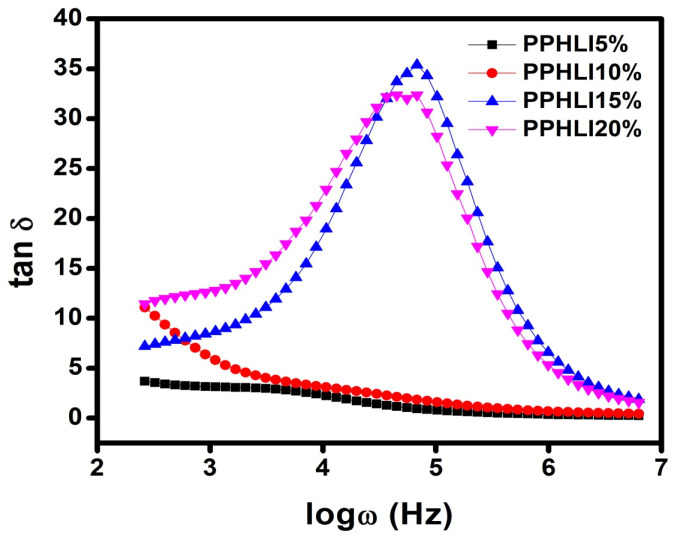
Tangent analysis for PPHLI5 wt.%, PPHLI10 wt.%, PPHLI15 wt.%, and PPHLI20 wt.% polymers.

**Figure 12 polymers-17-01758-f012:**
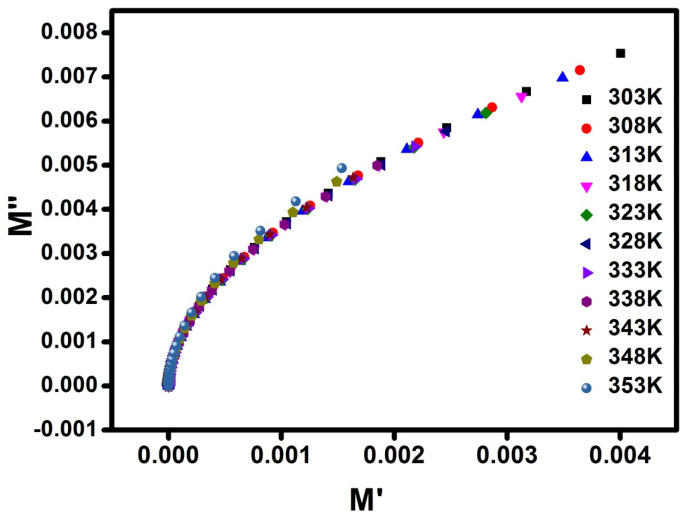
Argand plot for the PPHLI15 wt.% blend polymer electrolyte at various temperatures.

**Table 1 polymers-17-01758-t001:** Comparative data on conductivity, activation energy, dielectric constant, and relaxation time at different concentrations.

Sample	Conductivity (S·cm^−1^)	Activation Energy	Dielectric Constant	Relaxation Time (S)
PPHLI 5%	1.2385 × 10^−7^	−2.843 × 10^−9^	2.876 × 10^3^	2.517 × 10^−4^
PPHLI 10%	6.8678 × 10^−7^	−7.529 × 10^−7^	2.918 × 10^4^	3.942 × 10^−4^
PPHLI 15%	7.504 × 10^−5^	−6.824 × 10^−5^	1.586 × 10^6^	1.740 × 10^−4^
PPHLI 20%	2.837 × 10^−5^	−8.195 × 10^−6^	1.21173 × 10^5^	1.440 × 10^−4^

## Data Availability

Data is contained within the article.

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
