# Peer review of "Investigation into PVDF-HFP and PVP Polymer Blend Electrolytes with Lithium Ions for Energy Storage Application"

_polymers, 2025, doi:10.3390/polym17131758_

Round 1

Reviewer 1 Report

Comments and Suggestions for Authors

In the manuscript titled “Investigation of PVDF-HFP and PVP Polymer Blend Electrolytes Added with Lithium-Ion for Energy Storage Application”, the authors investigated the effect on ionic conductivity and dielectric properties of PVDF-HFP/PVP polymer electrolytes with different LiI additions, i.e. 5%, 10%, 15% and 20%. While the study presents some incremental improvements, the overall creativity and scientific significance remain modest. Additionally, the manuscript suffers from incomplete data representation and lacks a thorough discussion of the findings. Regrettably, in its present form, the work does not qualify for acceptance in Polymers. The following comments are provided to assist in strengthening the research.
1. There are many writing errors in the manuscript, such as KIO3 in lines 50 and 51, LiBF4 in lines 60 and 100, BF4 in line 66, and so on.
2. In Figure 1, the XRD patterns of pure PVDF-HFP and PVP are missing for comparison.
3. In Figure 2 and Figure 3, the location of the characteristic peaks and the groups they represent are not marked in the Figures, and the offset of the specific peaks mentioned in the text is not prominent, making it difficult for the reader to understand.
4. PVDF-HFP/PVP on line 187 and lithium iodide on line 221 are misspelled as PVDF-HEP/PVP and lithium nitrate, respectively.
5. The names of the samples in the manuscript are confused, e.g., PPHLI15%, PPHLI15 wt.%, PPHLI15 percent wt. Lithium Iodide and PVDF-HFP/PVP/15wt% Lithium iodide (Figure 5).
6. The plasticizing effects of EC are mentioned in line 254, but EC is not mentioned anywhere else in the text.
7. Figures 9a and b are plots of dielectric constant and dielectric loss vs. frequency. The author describes the relationship between the two and temperature. Similarly, the authors describe "With increasing temperature, the peak shifts to a higher frequency region as shown in figure, demonstrating the thermal relaxation process." in lines 295-297, which Figure shows that the peak shifts to high frequencies as the temperature increases?
8. The title is "Investigation of PVDF-HFP and PVP Polymer Blend Electrolytes Added with Lithium-Ion for Energy Storage Application", but there is a lack of relevant applications, such as electrochemical testing (cycle stability, rate capacity, etc) for assembled batteries.

Reviewer 2 Report

Comments and Suggestions for Authors

This manuscript studied the poly(vinylidene fluoride-co-hexafluoropropylene) (PVDF-HEP) and polyvinylpyrrolidone (PVP) blend polymers with lithium iodide as solid polymer electrolyte. The experiments are well organized and results discussion part is also reasonable. Therefore, it could be accepted by the journal of Polymers before addressing the follows issues.

  1. Four solid polymer electrolytes were prepared and many parameters, such as conductivity were studied. Among these four electrolytes, the PPHLI 15% showed the best performance according to the results. The authors should provide the deep expiations for the reasons of its superiority.
  2. It is better to label the represented vibrations in Figure 3.
  3. For temperature-dependent experiment, the temperature range is 303-373 K, however, only results in the range of 303 K to 353 K were discussed. What is the reason for this inconformity? In addition, the authors should provide experiments details for these testing, including the way to control the temperature and its accuracy.
  4. There are some formatting errors and grammar issues. The authors need to thoroughly review this work for any format and grammar issues.

Reviewer 3 Report

Comments and Suggestions for Authors

This manuscript presents a study on lithium-ion-conducting solid polymer electrolytes based on PVDF-HFP/PVP blends with varying lithium iodide (LiI) content. The authors investigate the structural, dielectric, and electrical properties using standard characterization methods such as XRD, FTIR, impedance spectroscopy, and dielectric modulus analysis. However, it requires major revision to improve scientific rigor, broaden the scope of characterization, and enhance clarity in both language and interpretation.

  1. Abbreviations must be clearly defined upon first introduction.
  2. SEM or AFM surface morphology should be presented to confirm the amorphous/crystalline distribution and film uniformity.
  3. Thermal analysis (e.g., TGA/DSC) and electrochemical window tests (e.g., CV/LSV) are lacking.
  4. A summary table comparing conductivity, activation energy, dielectric constant, and relaxation time among all samples would significantly enhance clarity.
  5. The role of Li⁺ transport mechanisms (segmental motion, hopping pathways) should be interpreted more profoundly with reference to the literature or molecular models.
  6. The influence of dielectric behavior and non-Debye relaxation on the practical device performance should be more comprehensively explained.
  7. Figure legends should specify the measurement conditions (e.g., temperature, frequency).
  8. The certain cycle performance in all solid-state batteries (e.g., constant current charge-discharge, Critical Current Density Tests) has not been addressed.
  9. It’s suggested to addperformance comparison tables (such as ion conductivity, ESW, mechanical strength, capacity performance, cycle stability, energy density) with recently published other PVDF-HFP with lithium salt based solid state polymer electrolyte articles.

Round 2

Reviewer 1 Report

Comments and Suggestions for Authors

No

Reviewer 3 Report

Comments and Suggestions for Authors

The author provided detailed responses to the questions I raised and revised the manuscript. I recommend this paper for publication in Polymers.